

# Effect of atrial fibrosis on clot burden score and physicochemical properties of thrombus in patients with ischaemic stroke occurring in non-valvular atrial fibrillation

Juan Zhao[1,2], Guangjun Deng[3], Weijing Wang[2], Peng Wang[2], Xinyu Shen[2], Xiaoxiao Yuan[2], Haifei Jiang[4] and Zhong-bao Ruan[1,5]

[1] Graduate School of Dalian Medical University, Dalian Medical University, Dalian, China
[2] Department of Cardiology, Tongzhou People's Hospital, Nantong, Jiangsu, China
[3] Department of Medical Imaging, Tongzhou People's Hospital, Nantong, Jiangsu, China
[4] Department of Cerebrovascular Interventional Diagnosis and Treatment Center, Tongzhou People's Hospital, Nantong, Jiangsu, China
[5] Department of Cardiology, The Affiliated Taizhou People's Hospital of Nanjing Medical University, Taizhou School of Clinical Medicine, Nanjing Medical University, Taizhou, Jiangsu, China

Corresponding author
Zhong-bao Ruan, tzcardiac@163.com

## ABSTRACT

**Background**. To investigate the effect of the degree of atrial fibrosis on the clot burden score (CBS) and physicochemical properties in patients with acute ischaemic stroke (AIS) due to non-valvular atrial fibrillation (NVAF).

**Methods**. A total of 117 patients with AIS in NVAF attending the Department of Cardiovascular Medicine and the Cerebrovascular Diagnostic and Treatment Centre between August 2021 and May 2024 were included in the study. Baseline clinical data, biochemical indexes, and imaging data of the patients were collected, and the patients were divided into 93 cases of the CBS (score of 0–6) group and 24 cases of the CBS (score of 7–10) group according to the CBS. CBS (score of 0–6) signifies higher clot burden. The enzyme-linked immunosorbent assay was used to measure the concentration of galactaglutinin-3 (gal-3) and transforming growth factor (TGF-β1) in the serum of the patients, and the PTFV1 were collected by 12-lead electrocardiogram, and the differences in the degree of atrial fibrosis between different groups and the risk factors of CBS (score of 0–6) were analysed. To analyse the effect of atrial fibrosis on the collateral circulation of stroke, the patients were divided into 31 cases with good collateral circulation (grade 3–4) and 86 cases with poor collateral circulation (grade 0–2) according to the digital subtraction angiography (DSA) images. The cerebral thrombus was collected from 60 AIS patients who underwent mechanical thrombectomy. The content of erythrocyte, fibrin/platelets and leukocytes in the thrombus was analysed by Mathew's scarlet blue staining, and the density of thrombus was measured by computed tomography (CT).

**Results**. A total of 117 patients were included in this study, and the proportion of hypertensive patients, proportion of chronic atrial fibrillation (CAF), B-type natriuretic peptide (BNP), neutrophil/lymphocyte ratio (NLR), D-dimer, uric acid concentration, proportion of patients with PTFV1 < −0.03 mm s, gal-3, and TGF-β1 were higher in the CBS (score of 0–6) group as compared to the CBS (score of 7–10) group (P-value
< 0.05). Hypertension, proportion of CAF, homocysteine, NLR, D-dimer, uric acid, PTFV1 < −0.03 mm s, gal-3, and TGF-β1, were risk factors for the development of high CBS in atrial fibrillation (AF), and hypertension and CAF were the most important factors for the occurrence of AF in the independent risk factors for stroke combined with high clot burden. gal-3 and TGF-β1 were risk factors for poor collateral circulation, atrial fibrosis indexes were not associated with thrombus pathological composition and thrombus density.

**Conclusions.** Atrial fibrosis increases clot burden in patients with AIS due to NVAF but does not significantly correlate with the physicochemical properties and density of the thrombus.

## INTRODUCTION

Atrial fibrillation (AF) is one of the most common clinical arrhythmias, and atrial fibrosis is one of the most critical substrates for the occurrence and maintenance of AF which can result in heart failure (HF) and acute ischaemic stroke (AIS). Atrial fibrosis is caused by an imbalance in the production and degradation of extracellular matrix, especially the excessive deposition of collagen, which in turn leads to the formation of scar tissue in the intercellular matrix. Atrial fibrosis causes blood to stagnate in the atria by altering the intra-atrial hemodynamics, promoting platelet aggregation and coagulation, and facilitating thrombosis (*Ronsoni et al., 2021*). One of the most common complications is stroke, and it has been shown that patients with AF are at about five times the risk of stroke as patients without AF, and that 20 percent of strokes are caused by AF. Even though the incidence of stroke has been significantly reduced by oral anticoagulation, AF is thought to account for a significant proportion of cryptogenic strokes where no etiology is identified (*Elsheikh et al., 2024*). Growing evidence suggests a consistent association between AF and cognitive impairment and dementia, as AF increases the risk of cerebral hypoperfusion, inflammation, and stroke. The prevalence of dementia was more than double in those with compared to those without AF (*Rivard et al., 2022*; *Kim, Yang & Joung, 2021*). Recent studies have shown that the higher the degree of atrial fibrosis, the more likely it is to cause endocardial endothelial injury, and the worse the prognosis (*Miyauchi et al., 2023*). Clot burden score (CBS), collateral circulation status, the percentage and density of thrombus components can all affect the postoperative prognosis of patients with stroke (*Derraz et al., 2019*; *Sinha, Gupta & Bhaskar, 2024*; *Fereidoonnezhad et al., 2021*). Currently, there are fewer reports on the effects of AF on the clot burden and the physicochemical properties of thrombi in patients with non-valvular atrial fibrillation (NVAF) combined with AIS. In this study, we retrospectively analyzed the CBS, collateral circulation, and thrombus density of patients with AIS due to NVAF based on computed tomography (CT) angiography (CTA), and investigated the influence of atrial fibrosis on CBS and thrombotic properties affecting prognosis of stroke patients.

## MATERIALS & METHODS

### Study subjects

The patients with AF-associated cardiogenic cerebral infarction were selected according to the criteria of the TOAST classification in this study. 117 patients with AIS occurring in NVAF who attended the Department of Cardiovascular Medicine and Cerebrovascular Diagnostic and Treatment Centre of Tongzhou People's Hospital in Nantong City from August 2021 to May 2024 were included. The study was approved by ethics board of Tongzhou People's Hospital (2021-K012) and informed consent was taken from all the patients, written informed consent was obtained.

Patients were included if they satisfied the following: (I) age ≥18 years combined with AF; according to the 2020 European Society of Cardiology Guidelines for AF (*SEC Working Group for the 2020 ESC Guidelines for the Management of Atrial Fibrillation et al., 2021*), the diagnosis of AF needs to be based on a transmural electrocardiogram (ECG) or an ambulatory ECG recording of a duration longer than 30s and the presence of an ECG waveform of AF in the recording. (II) The presence of intracranial anterior circulation macrovascular occlusion (including the intracranial segment of the internal carotid artery, the middle cerebral artery M1 and M2 segments, and the anterior cerebral artery A1 segment) was clearly identified by CTA after admission to the hospital. (III) The baseline NIHSS score was a large ≥ 6 points, and the modified mRs score before the onset of the disease was < 2 points.

Patients were excluded if they presented with the following: (I) Intracranial haemorrhagic disease; active bleeding or significant bleeding tendency. (II) Previous catheter ablation of AF or percutaneous left atrial appendage occlusion. (III) Rheumatic heart valve disease, valvular AF (moderate-severe mitral stenosis, mechanical valves). (IV) Incomplete follow-up data. (V) Complicated with severe other diseases including vital organ failure and malignant tumours.

### CBS criteria

CTA-based CBS method was used, with a total score of 10 points for CBS, and different values were assigned to the vessels of the site, as follows: 1 point for each of the lower section of the internal carotid artery of the bedrock protuberance, the upper trunk of the M2 of the middle cerebra artery (MCA), the lower trunk of the M2 of the MCA, and the A1 section of the anterior cerebral artery, and 2 points for each of the upper section of the internal carotid artery of the bedrock protuberance, the proximal and distal sections of the M1 of the MCA, and if there were any occlusion, the corresponding points would be subtracted. Lower scores represent higher clot burden (*Li et al., 2024*; *Menon et al., 2018*). Patients were divided into CBS (score of 0–6) group and CBS (score of 7–10) according to CBS criteria.

### Collateral circulation scoring criteria

Assessment of collateral circulation: pial collateral vessels were evaluated based on DSA (*Higashida et al., 2003*; *Bates et al., 2007*).

 

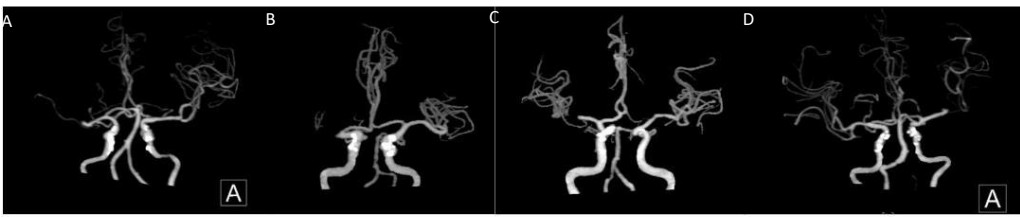

**Figure 1  Middle cerebral artery occlusion.** (A, B) Poor compensation of collateral circulation. (C, D) Well compensated by collateral circulation.

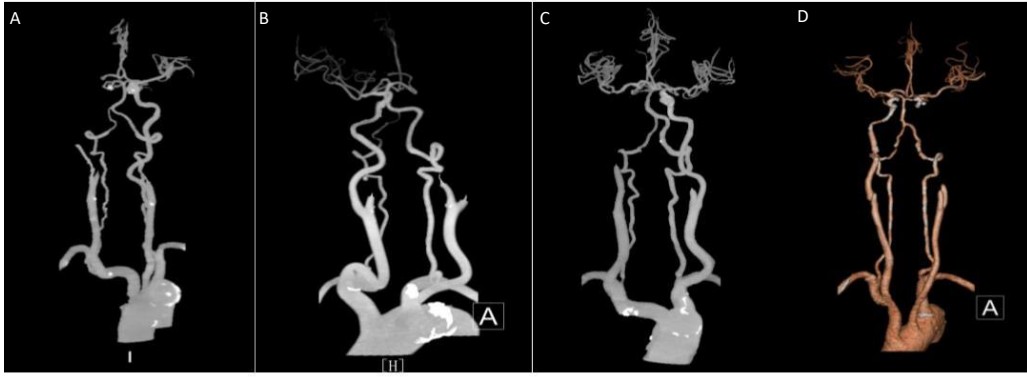

**Figure 2  Internal carotid artery.** (A, B) Poor compensation of collateral circulation. (C, D) Well compensation of collateral circulation.

ASITN/SIR rating

| Items | Standard of scoring |
|---|---|
| Grade 0 | No collateral vessels on the ischaemic side |
| Grade 1 | Little blood flow in the peripheral collateral branches on the ischaemic side and no blood flow in some areas |
| Grade 2 | A lot of blood flow in the peripheral collateral branches on the ischaemic side, but only partial blood flow in the ischaemic focus |
| Grade 3 | Low but complete blood flow in the late venous stage |
| Grade 4 | Collateral blood supplies the entire vascular area |

ASITN/SIR: American Society of Interventional and Therapeutic Neuroradiology/Society of Interventional Radiology.

Based on this criterion, two grades of 0–2 are further classified as poor collateral circulation and 3–4 as good collateral circulation (Figs. 1 and 2).

Clinical data: The patient's gender, age, history of previous chronic diseases (*e.g.*, hypertension, diabetes mellitus, HF), B-type natriuretic peptide (BNP), neutrophils, lymphocytes, plasminogen, fibrinogen, D-dimer, uric acid (UA), and other clinical data were collected.

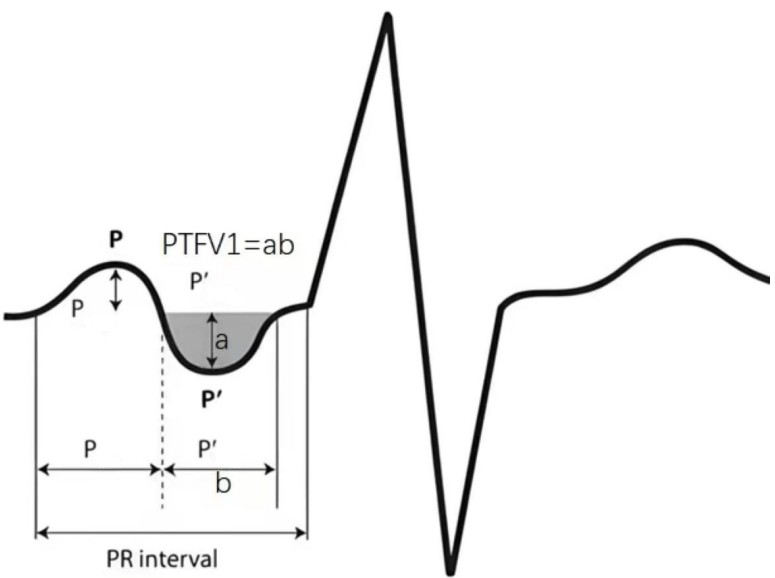

**Figure 3** **The product of the voltage (mm) and the time limit (s) of the negative P-wave was taken as the PTFV1 value.**

Biochemical indexes: Three ml of fasting venous blood was collected on the first day of admission, centrifuged at 1,500*g for 20 min, and the upper layer of serum was collected and frozen in −80 °C refrigerator for spare use, and enzyme-linked immunosorbent assay was used to determine the content of gal-3 and TGF-β1.

Imaging data: Colour Philips ultrasonography was applied to determine the left atrial end-diastolic internal diameter, left ventricular end-systolic internal diameter, and left ventricular ejection fraction. All elderly subjects underwent a routine 12-lead ECG scan in lead V1, and the product of the voltage (mm) and the time limit (s) of the negative P-wave was taken as the PTFV1 value (Fig. 3).

Pathological accumulation of thrombus: The extracted cerebral thrombus was promptly fixed in 4% paraformaldehyde for 12 h and subsequently embedded in paraffin. Thrombus tissue was dehydrated in ethanol of varying concentration gradients. The paraffin sections were dewaxed until rehydrated, and conventional paraffin sections were immersed in MBS staining solution for staining, then rinsed with distilled water, and the slides were sealed after drying. The Image J software was employed for quantitative analysis of the photographed tissue sections. Image J software is a segmentation and analysis software based on computational color, capable of obtaining quantitative information of erythrocyte, fibrin, and leukocyte constituting the thrombus.

## Thrombus density measurement

The measurement of thrombus density was carried out based on a five mm head CT scan to delineate a circular region of interest (ROI) within the thrombus (the size of which accounted for approximately 2/3 of the vascular area) (Fig. 4). The mean CT value of the ROI region was recorded as the absolute value of thrombus density (aHU), and in the same

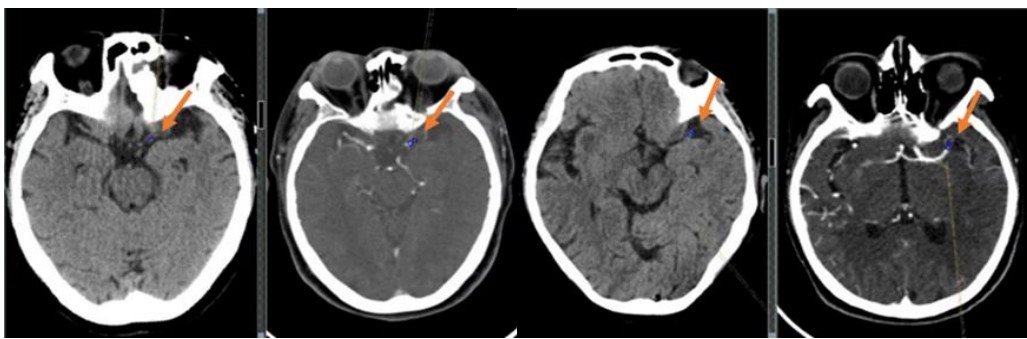

**Figure 4** **Examples of clot perviousness measurement.** Contrast-enhanced CT (CECT) showed occlusion in the segment of left middle cerebral artery.

way, the mean CT value of the symmetrical vessel on the healthy side was recorded as the mean CT value of the symmetrical vessel on the healthy side (cHU). The relative value of thrombus density was recorded as the difference (dHU = aHU-cHU) or ratio (rHU = aHU/cHU) between the affected and healthy sides (*Ye et al., 2021*).

## Statistical analysis
All data were analyzed using SPSS 26.0 software, the normality test was used for the measurement data, which conformed to the normal distribution was expressed by x ± s, and the comparison between two groups was performed by the independent samples $t$-test, and the measurement data that did not conform to the normal distribution was described by the M(Q1,Q3), and the comparison between two groups was performed by the Mann–Whitney $U$ test, the comparison between two groups was performed by the non-parametric test, and the comparison between the categorical variables was performed by the chi-square test. Non-parametric test was used for comparison, and chi-square test was used for comparison between two groups for categorical variables. Risk factors were analyzed by logistic regression analysis, and then risk factors were included in multifactorial analysis, and the difference was considered statistically significant at $P < 0.05$. Receiver operating characteristic (ROC) curve analysis was used to assess the diagnostic value of atrial fibrosis indicators in predicting high clot burden in stroke, and linear regression was used to analyze the relationship between atrial fibrosis and the physicochemical properties and density of thrombus.

## RESULTS
### Baseline data of patients
A total of 117 patients with stroke caused by AF were included in this study, including 69 males (59%), age 75 (67,80) years, 24 cases with low clot burden and 93 cases with high clot burden. General data were compared between the two groups (Table 1): the percentage of patients with hypertension, CAF, BNP, NLR, D-dimer, uric acid concentration in the CBS (score of 0–6) group, percentage of patients with PTFV1 < −0.03 mm s, gal-3, and TGF-β1 were higher in the CBS (score of 0–6) group compared with the CBS (score of 7–10) group

**Table 1  Comparison in clinical data between CBS (score of 7–10) group and CBS (score of 0–6) group.**

| Characteristics | CBS (score of 7–10) group ($n = 24$) | CBS (score of 0–6) group ($n = 93$) | All patients ($n = 117$) | *P* value |
|---|---|---|---|---|
| Age, y | 74 (67, 79) | 76 (67.5, 80) | 75 (67, 80) | 0.561 |
| Sex, male | 14 (58.3) | 55 (59.1) | 69 (59.0) | 0.943 |
| Hypertension | 11 (45.8) | 65 (69.9) | 76 (65.0) | 0.028 |
| Diabetes | 6 (25) | 33 (35.5) | 39 (33.3) | 0.331 |
| Coronary artery disease | 7 (29.2) | 34 (36.6) | 41 (35.0) | 0.499 |
| heart failure | 14 (58.3) | 46 (49.5) | 60 (51.3) | 0.438 |
| Type of occluded vessels | | | | 0.296 |
| Internal carotid artery | 8 (33.3) | 42 (46.2) | 50 (42.7) | |
| Middle cerebral artery | 16 (66.6) | 51 (54.8) | 67 (57.3) | |
| Type of AF | | | | 0.014 |
| Paroxysmal AF | 16 (66.6) | 36 (38.8) | 52 (44.4) | |
| CAF | 8 (33.3) | 57 (61.3) | 65 (55.6) | |
| BNP | 717.1 (436, 1,302.5) | 1,333 (657.4, 2,102) | 1,232 (645.5, 1,862.64) | 0.01 |
| Homocysteine | 12.32 (6.91, 17.67) | 14.82 (9.81, 23.33) | 14 (9.29, 22.54) | 0.054 |
| NLR | 3.74 (3.02, 5.3) | 4.86 (3.43, 8.105) | 4.72 (3.24, 7.49) | 0.0490 |
| Prothrombin time | 12.2 (11.45, 13.33) | 11.900 (11.5, 12.55) | 12 (11.5, 12.75) | 0.166 |
| Fibrinogen | 2.79 (2.07, 3.95) | 2.72 (2.35, 3.37) | 2.72 (2.31, 3.42) | 0.769 |
| D-dimer | 0.65 (0.29, 1.8450) | 2.08 (0.54, 3.24) | 1.60 (0.49, 3.06) | 0.021 |
| UA | 315.5 (253.36, 395.56) | 386.5 (300.7, 513.1) | 367.8 (288.70, 494.55) | 0.01 |
| Ventricular ejection fraction | 0.6 (0.54, 0.63) | 0.59 (0.55, 0.64) | 0.59 (0.55,0.64) | 0.831 |
| End-diastolic internal diameter | 44.00 (42.25, 48.25) | 46.00 (43, 48) | 46 (43,48) | 0.406 |
| End-systolic internal diameter | 32.00 (28.5, 34.00) | 31.00 (29, 33.00) | 31 (29,33.5) | 0.316 |
| PTFV1<-0.03 mm s | 6 (25) | 45 (48.39) | | 0.040 |
| gal-3 | 58.48 (46.11, 66.15) | 68.53 (62.62, 75.14) | 66.6 (57.54, 74.03) | 0.001 |
| TGF-β1 | 32.31 ± 1.82 | 40.87 ± 1.00 | 39.11 ± 10.04 | 0.000 |
| CHA2DS2-VASc | 5 (4.5, 6) | 5 (4, 6) | 6 (5, 7) | 0.909 |

(*P*-value < 0.05). There was no statistically significant difference between the two groups in terms of age, gender, past history including diabetes, coronary artery disease, HF, occluded vessels, prothrombin time, fibrinogen, ventricular ejection fraction, end-diastolic internal diameter, and end-systolic internal diameter (*P*-value > 0.05).

Logistic regression analysis of the relationship between CBS (score of 7–10) group and CBS (score of 0–6) group (Table 2). Univariate logistic regression analysis concluded that hypertension, proportion of CAF, homocysteine, NLR, D-dimer, uric acid, PTFV1 < −0.03, gal-3, TGF-β1 were risk factors for stroke and CBS (score of 0–6) group in AF. Multivariate regression analysis of variables with *P*-value<0.05 in univariate analysis suggested that hypertension and CAF were independent risk factors for stroke with CBS (score of 0–6) group in AF.

Patients with AIS due to AF were grouped by CBS, and the predictive value of CBS (score of 0–6) group was evaluated by ROC curve (Fig. 5, Table 3). For single detection, the cut-off value of gal-3 was 67.128 (AUC = 0.723) and the cut-off value of TGF-β1 was

**Table 2 Logistic regression analysis of the relationship between CBS (score of 7–10) group and CBS (score of 0–6) group.**

| Variables | Univariate | | | Multivariate | | |
|---|---|---|---|---|---|---|
| | OR | p | 95% CI | OR | p | 95% CI |
| Age, y | 1.006 | 0.803 | 0.959–1.055 | | | |
| Sex, male | 0.967 | 0.943 | 0.389–2.405 | | | |
| Hypertension | 2.744 | 0.031 | 1.097–6.864 | 4.409 | 0.034 | 1.119–17.377 |
| Diabetes | 0.606 | 0.334 | 0.219–1.676 | | | |
| Coronary artery disease | 1.4 | 0.5 | 0.527–3.715 | | | |
| HF | 0.699 | 0.44 | 0.282–1.733 | | | |
| CAF | 3.167 | 0.017 | 1.230–8.153 | 8.694 | 0.003 | 2.086–36.229 |
| Homocysteine | 1.068 | 0.05 | 1–1.14 | | | |
| NLR | 1.201 | 0.049 | 1.001–1.440 | | | |
| Prothrombin time | 0.983 | 0.749 | 0.885–1.092 | | | |
| Fibrinogen | 0.8 | 0.449 | 0.449–1.425 | | | |
| D-dimer | 1.396 | 0.048 | 1.003–1.943 | | | |
| UA | 1.005 | 0.016 | 1.001–1.009 | | | |
| Ventricular ejection fraction | 0.812 | 0.951 | 0.001–653.5 | | | |
| End-diastolic internal diameter | 1.013 | 0.793 | 0.919–1.177 | | | |
| End-systolic internal diameter | 0.971 | 0.54 | 0.885–1.066 | | | |
| PTFV1< −0.03 mm s | 2.812 | 0.045 | 1.025–7.718 | | | |
| gal-3 | 1.08 | 0.001 | 1.033–1.128 | | | |
| TGF-β1 | 1.104 | 0.000 | 1.045–1.166 | | | |

31.743 (AUC = 0.737). For combined detection, the diagnostic efficiency was the highest (AUC = 0.767), the sensitivity was 0.72, the specificity was 0.792, and the Jordon index was 0.512.

Logistic regression analysis of factors influencing lateral circulation (Table 4).

In univariate logistic regression analysis with left ventricular ejection fraction, end-diastolic internal diameter, end-systolic internal diameter, PTFV1 < −0.03 mm s, gal-3, and TGF-β1 as independent variables, gal-3, and TGF-β1 were the risk factors for poor collateral circulation for stroke due to AF, and then included in the multifactorial logistic regression analysis, and neither was an independent risk factor.

The cerebral thrombus was collected from 60 AIS patients who underwent mechanical thrombectomy. The content of erythrocyte, fibrin/platelets and leukocytes in the thrombus was analysed by Mathew's scarlet blue staining. PTFV1 < −0.03 mm s, gal-3, TGF-β1, ejection fraction, end-diastolic diameter, end-systolic diameter were taken as independent variables, erythrocyte percentage (Figs. 6–7), dHU, rHU and aHU were taken as dependent variables. Linear regression analysis showed that erythrocyte percentage and thrombosis density were not related to PTFV1 < −0.03 mm s, gal-3, TGF-β1, ejection fraction, end-diastolic diameter, end-systolic diameter (Table 5).

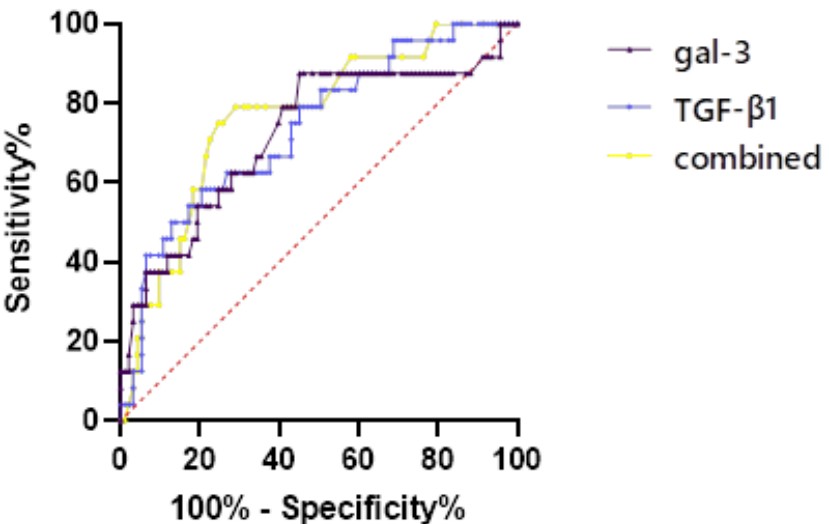

**Figure 5** The predictive value of CBS (score of 0–6) group was evaluated by ROC curve.

**Table 3** The predictive value of CBS (score of 0–6) group was evaluated by ROC curve.

| Variables | Optimal cut-off value | AUC | Standard error | 95% CI | Jordon index | Sensitivity | Specificity |
|---|---|---|---|---|---|---|---|
| gal-3 | 67.128 | 0.723 | 0.064 | 0.597–0.848 | 0.423 | 0.548 | 0.875 |
| TGF-β1 | 31.743 | 0.737 | 0.057 | 0.625–0.848 | 0.379 | 0.796 | 0.583 |
| Combined | 0.798 | 0.767 | 0.053 | 0.661–0.870 | 0.512 | 0.72 | 0.792 |

**Table 4** Logistic regression analysis of factors influencing lateral circulation.

| | univariate | | |
|---|---|---|---|
| Variables | OR | p | 95% CI |
| Ventricular ejection fraction | 0.012 | 0.195 | 0–9.636 |
| End-diastolic internal diameter | 0.957 | 0.344 | 0.873–1.049 |
| End-systolic internal diameter | 0.985 | 0.737 | 0.903–1.075 |
| PTFV1<−0.03 mm s | 1.581 | 0.29 | 0.676–3.695 |
| Gal-3 | 1.059 | 0.004 | 1.018–1.101 |
| TGF-β1 | 1.062 | 0.009 | 1.015–1.110 |

## DISCUSSION

Stroke caused by AF is usually fatal or results in severe disability (*Ozdemir et al., 2023*). PTFV1 is an important indicator of diffuse left atrial stromal fibrosis, suggesting abnormalities in atrial structure or function. When left atrial burden is increased and fibrosis occurs, the left atrial depolarisation time is prolonged and the depolarisation vector is increased, and these changes may manifest themselves on the electrocardiogram as a negatively increased PTFV1 value (*Nakatani et al., 2019*; *Ikenouchi et al., 2023*). Gal-3

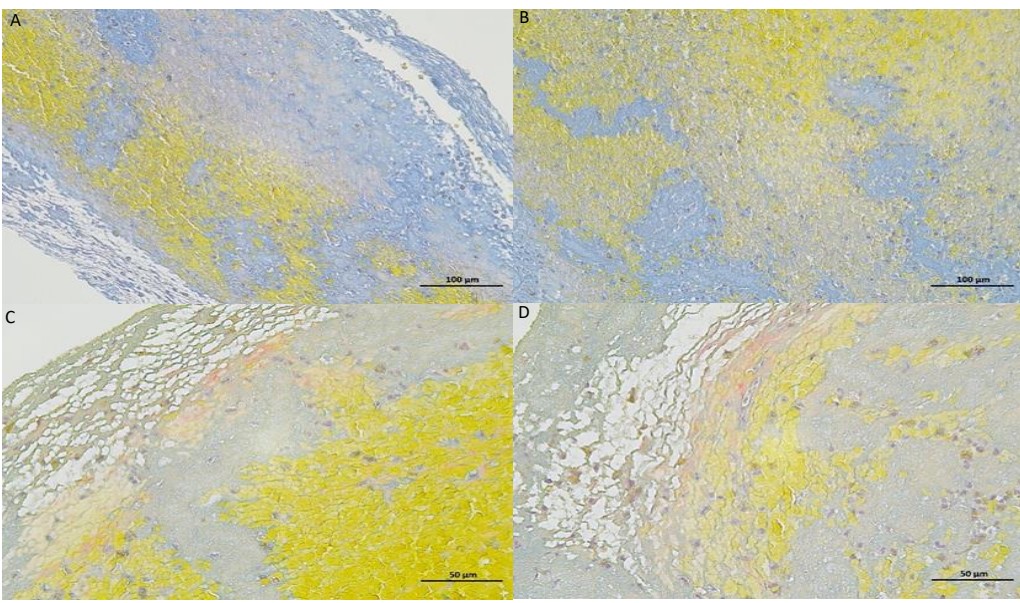

**Figure 6** **Thrombus MSB stained section diagrams.** The yellow areas are red blood cells, the blue areas are collagen fibers, and the gray areas are platelets.

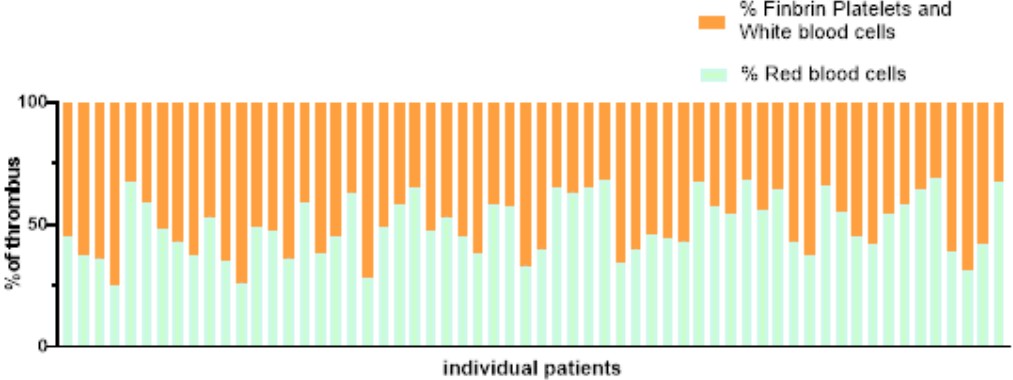

**Figure 7** **Proportion of blood cell composition in 60 patients.**

acts to activate the pro-fibrotic phenotype of macrophages, promotes migration and proliferation of fibroblasts or myofibroblasts, participates in angiogenesis, mediates neutrophil adhesion and recruitment to injured myocardium, and promotes the production of extracellular matrix collagen and ventricular remodelling (*Seropian et al., 2023*; *Ma, Chen & Ma, 2021*). TGF-β1 in the TGF-β1/Smad signalling pathway can stimulate the differentiation of atrial fibroblasts through the Smad signalling pathway, promoting cell migration, proliferation and extracellular matrix protein synthesis, which can cause collagen deposition and lead to myocardial fibrosis. Negative values of PTFV1, the expression of gal-3, and TGF-β1 in atrial tissues are positively correlated with the degree of AF. Left

**Table 5  Multiple linear regression analysis of atrial fibrosis indexes and physicochemical properties of thrombus (proportion of blood cell composition and thrombus density).**

| | Erythrocyte percentage | | dHU | | rHU | | aHU | |
|---|---|---|---|---|---|---|---|---|
| | β | *P* value | β | *P* value | β | *P* value | β | *P* value |
| PTFV1 <−0.03 mm s | 0.012 | 0.937 | 0.37 | 0.805 | 0.03 | 0.840 | −0.018 | 0.890 |
| gal-3 | 0.264 | 0.188 | −0.128 | 0.523 | −0.229 | 0.246 | 0.296 | 0.091 |
| TGF-β1 | −0.108 | 0.582 | 0.134 | 0.497 | 0.048 | 0.803 | 0.275 | 0.110 |
| Ventricular ejection fraction | −0.132 | 0.389 | 0.009 | 0.952 | −0.004 | 0.977 | 0.016 | 0.905 |
| End-diastolic internal diameter | 0.123 | 0.594 | 0.160 | 0.490 | 0.240 | 0.293 | −0.160 | 0.425 |
| End-systolic internal diameter | −0.256 | 0.294 | 0.043 | 0.860 | −0.02 | 0.933 | 0.207 | 0.329 |

atrial strain (LASr) represents a relatively new but promising technique for left atrial and left ventricle function evaluation. LASr was strongly linked to myocardial fibrosis and endocardial thickness. The lower the LASr magnitude, the greater the extent of myocardial fibrosis on cardiac magnetic resonance imaging (*Kuppahally et al., 2010*; *Tanasa et al., 2021*). The degree of atrial fibrosis >20% significantly increases the risk of stroke, reduces atrial blood flow velocity, and makes it easier for blood to stagnate in the auricle, leading to thrombosis, suggesting that the degree of atrial fibrosis is positively correlated with left auricle thrombosis. Recent studies have shown that the degree of atrial fibrosis is significantly associated with the risk and severity of stroke (*Mahnkopf, Kwon & Akoum, 2021*). This correlation is mainly reflected in the effect of atrial fibrosis on the clot burden, and studies have shown that the more severe the atrial fibrosis, the higher the clot burden in stroke may be. The mechanism for this phenomenon primarily involves a reduction in the contractile function of the atria, leading to haemodynamic changes that increase the risk of thrombus formation. Recent evidence indicates that LASr impairment in AF patients may noninvasively predict left atrial appendage dysfunction and thrombosis. Speckle tracking echocardiography examination should be considered for implementation in the clinical practice, due to its incremental diagnostic and prognostic value over conventional transthoracic echocardiography. But it has not been popularized in clinical work (*Kupczynska et al., 2017*; *Sonaglioni et al., 2021*). In addition, electrophysiological changes and structural remodelling of the atria during AF provide favourable conditions for thrombus formation, and abnormal electrical activity within the atria promotes platelet aggregation, further exacerbating the clot burden (*Escudero-Martínez, Morales-Caba & Segura, 2023*). It has been suggested, in support of the hypothesis that atrial fibrosis can be independent of AF in leading to thrombosis and cardiac stroke, that fibrosis is caused by a mechanism of structural and electrical remodelling of the atria, cardiac fibroblasts, and mechanical-functional coupling. The biophysical relationship between them plays an integral role in the physicochemical properties of thrombi (*Tandon et al., 2019*). In this study, we found significant differences in atrial fibrosis indexes between different cloth burden groups as an influencing factor for high clot burden, which is consistent with previous findings. In addition, we concluded using ROC curves that gal-3 and TGF-β1 were

effective in predicting CBS (score of 7–10) in patients with stroke, and the combination of the two has a higher predictive diagnostic value.

The proportion of hypertensive patients and CAF patients, BNP, NLR, UA, D-dimer, concentration, percentage of PTFV1 < −0.03 mm s, gal-3, and TGF-β1 were higher in the CBS (score of 0–6) group, and hypertension and CAF were independent risk factors for high clot burden. From the point of view of the effect of myocardial fibrosis on high clot burden this may be attributed to the fact that hypertension exacerbates diffuse myocardial fibrosis by increasing cardiac burden, damage to cardiomyocytes, interstitial fibrosis, and alterations in cardiac function, electrical activity, and perfusion resulting in enlargement of the left atrium and generation of more refractory waves (*González et al., 2024*). Microcirculation ischemia caused by atrial fibrosis is one of the important predictors of myocardial infarction. As the atrial fibrosis worsens, the prevalence of coronary heart disease in AF patients is estimated to be 3–4 times higher than that in the general population. which is closely related to poor heart rate control, acute presentation with hemodynamic collapse and pulmonary oedema, increased likelihood of bleeding complications and poor response to ablation therapies (*Batta, Hatwal & Sharma, 2024*; *Mekhael et al., 2024*).

According to the duration and clinical manifestations, AF is mainly divided into four types: initial atrial fibrillation, paroxysmal atrial fibrillation, persistent atrial fibrillation, and permanent atrial fibrillation. The longer the duration of AF in patients with atrial fibrillation, the more severe myocardial fibrosis (*Li et al., 2022*). The proportion of PTFV1 < −0.03 mm s, gal-3 and TGF-β1 have effects on clot burden, which further indicates that atrial fibrosis can affect the occurrence of high clot burden in stroke patients with NVAF. Early individualized evaluation of atrial fibrosis in patients with AF, the application of drugs to improve ventricular remodeling and delay myocardial fibrosis, in order to maximize the effect of interventional therapy and prognosis of stroke.

BNP is important in the pathogenesis, diagnosis, treatment, and prognosis of cardiovascular diseases. Elevated BNP levels in patients with AF correlate with the severity of atrial fibrosis detected on imaging, and BNP is elevated as a result of cardiogenic embolism, which improves prediction of the risk of thromboembolism (*Fonseca & Coelho, 2021*). It is now well established that elevated serum BNP correlates with increased post-stroke mortality and poor outcome (*Harpaz et al., 2020*). Inflammatory response plays an important role in stroke, and increased neutrophil counts correlate with the degree of inflammation in stroke. Increased neutrophil counts are associated with the degree of inflammation in stroke. After stroke, neutrophils rapidly accumulate around the lesion, releasing matrix metalloproteinase-9 to accumulate at the lesion site and cause secondary brain damage. AIS injury induces apoptosis and functional inactivation of lymphocytes, and several studies have shown that lymphocytes can repair inflammation-induced damage, that the NLR reflects the balance of the neutrophil-lymphocyte relationship, and that a higher NLR is associated with a poorer functional outcome at 3 months post-stroke, and that the NLR is a readily available and inexpensive test that can be used as a predictor of prognosis for patients with stroke (*Wan et al., 2020*).

In recent years, some studies have shown that Hs-CRP is closely related to the occurrence, development and prognosis of AF. Hs-CRP levels are not only associated with the overall

prognosis of AF, but may also play a key role in the prognosis of stroke caused by AF. High levels of hs-CRP may indicate a more severe inflammatory response and blood vessel damage, which may lead to more severe damage to brain tissue and more difficult recovery of nerve function after stroke. Therefore, monitoring Hs-CRP levels in clinical work may help us better assess stroke outcomes in patients with AF (*Zietz et al., 2024*).

Serum homocysteine Hcy, which leads to lethal reactive oxygen species (ROS) and lipid peroxidation, is not only a risk factor for stroke severity, poor prognosis, and stroke recurrence, but is also associated with stroke recurrence (*Amini et al., 2022*; *Lan et al., 2022*). D-dimer is representative of the concentration of total fibrin, and can therefore be used as a biomarker of intravascular fibrinolysis and thrombosis, and elevated plasma fibrinogen and D-dimer levels are associated with blood–brain barrier damage. Elevated levels of plasma fibrinogen and D-dimer are associated with blood–brain barrier damage and can enhance neuroinflammation and exacerbate neurological damage by promoting cytokine secretion by microglial cells and leukocyte recruitment, leading to increased mortality and poor prognosis in stroke (*Bao et al., 2023*). Hypercoagulability itself promotes atrial fibrosis. In rat atrial fibroblasts, thrombin increased the phosphorylation of the pro-fibrotic signaling molecule protein kinase B and extracellular regulated protein kinase, which resulted in a 2.7-fold increase in the expression of TGF-β1, a 6.1-fold increase in the expression of monocyte chemotactic protein-1, and a 2.5-fold increase in the synthesis of fibroblast collagen, all of which were inhibited by the thrombin inhibitor dabigatran. These pro-fibrotic effects were inhibited by the thrombin inhibitor dabigatran, suggesting that anticoagulation may not only prevent stroke but also atrial fibrosis. UA is a powerful antioxidant that scavenges ROS and protects cells from oxidative stress. Appropriate concentrations of UA are neuroprotective, so the extent of brain damage and ROS production are reduced with the addition of appropriate amounts of UA. Higher levels of UA proliferate the smooth muscle wall, enhance low-density lipoprotein oxidation, reduce endothelial nitric oxide synthase, which contributes to endothelial dysfunction, and increase platelet-derived growth factor production. Each of these factors may stimulate cascade coagulation leading to thrombosis and arterial occlusion. Experimental studies have shown that hyperuricaemia leads to elevated levels of systemic inflammatory factors and may also have the ability to induce systemic inflammation *via* the NF-kB signaling pathway, and that UA induces inflammation *via* the AMP-activated protein kinase-mTOR (mammalian target of rapamycin) mitochondrial ROS and hypoxia-inducible factor-1α pathways (*Zhang et al., 2023*). The paradoxical properties of UA may explain previous inconsistent findings on the effect of serum UA levels on the prognosis of inconsistent findings on the prognostic impact of AIS, serum UA levels are nonlinearly correlated with the prognosis of AIS (*Zhu et al., 2022*). In the present study, we collected clinical data from patients and analyzed the risk factors for high thrombotic burden in stroke, which were in general agreement with previous studies. In the course of clinical management, we should be alert to the role of general clinical data on the risk and prognosis of stroke occurrence due to AF in patients with NVAF, in addition to the adverse effects of AF on stroke.

Inflammation, oxidative stress, metabolic abnormalities and other factors can not only promote the occurrence and development of atrial fibrosis, but also indirectly lead to

myocardial ischemia through affecting the structure and function of coronary arteries. Atrial fibrosis is not only a key factor in the occurrence and persistence of atrial fibrillation, but also an independent predictor of poor clinical outcomes, including coronary heart disease, myocardial infarction, stroke, heart failure, and recurrence of AF after ablation (*Dilaveris et al., 2019*).

Collateral balance can predict the rate of cerebral infarction progression, the degree of recanalization, the likelihood of bleeding transformation, and various treatment opportunities (*Fukuda & Liebeskind, 2023*). The extent of intravascular thrombus and the quality of collateral filling in angiography can predict the clinical outcome of acute stroke patients. In this study, factors affecting collateral circulation were analyzed. gal-3 and TGF-β1 both affected the classification of collateral circulation. The decrease in atrial function caused by atrial fibrosis may affect the pumping efficiency of the heart, and then cause changes in cerebral hemodynamics. This change affects the opening of collateral circulation and blood perfusion, making it more difficult to obtain adequate blood supply to the ischemic area. In addition, AF works by increasing the release of vasoactive peptides such as angiotensin II or endothelin. Activation of this system not only promotes the process of myocardial fibrosis, but also may release a variety of inflammatory factors, such as cytokines and chemokines, through increased oxidative stress and inflammation of the blood vessel wall. These inflammatory factors can activate the inflammatory response of the blood vessel wall, attract more inflammatory cells (such as white blood cells, macrophages, *etc.*) to infiltrate the blood vessel wall, affect the establishment of collateral circulation and compensatory perfusion of tissue (*Dobrev et al., 2023*).

The structure of emboli mostly consists of erythrocytes, fibrinogen, platelets and leukocytes. We found that erythrocytes and fibrin occupied the most area by observing the structure of the thrombus. Therefore, erythrocyte-rich thrombus or fibrin-rich thrombus are commonly used to describe the tissue composition of the embolus and to group them. The composition of the thrombus in patients with cardiac stroke was dominated by fibrin/platelets, whereas the composition of the thrombus in non-cardiac stroke was dominated by erythrocytes (*Desilles et al., 2022*). However, over the years, scientists from various countries have conducted studies related to the aetiology of thrombosis, and the findings are not entirely consistent. Relevant studies have concluded that a high erythrocyte component is more common in patients with cardiac stroke, and some studies have suggested that the composition of the thrombus is not related to stroke aetiology. Relative thrombus density is positively correlated with erythrocyte fraction and significantly negatively correlated with fibrin/platelet content (*Ye et al., 2021*). Based on this study, we concluded that there is no linear correlation between emboli composition and thrombus density and the degree of atrial fibrosis in AF-induced cardiogenic cerebral infarction. The results of this study did not show statistical differences which may be limited by our sample size and further studies are needed to provide evidence. Reports on the correlation of thrombus density or thrombus composition with atrial fibrosis are still scarce and of low quality, and further studies with large sample sizes are needed.

There are some limitations in this study. Firstly, endomyocardial biopsy and fibrosis staining are currently the gold standard for diagnosing myocardial fibrosis, due to the

limitation of the conditions, this experiment only used multiple serum fibrosis indexes and PTFVI to analyze the relationship with the degree of AF myocardium fibrosis, and it is still necessary to use endomyocardial biopsy and fibrosis staining to assess the degree of cardiac fibrosis to further improve the present study in the subsequent studies. Secondly, this study was a single-centre retrospective study with a small number of patients enrolled, which may cause data bias in the statistical analysis. More multi-centre, large-sample randomized clinical controlled trials with uniform standards are still needed to explore the relationship in the future. Thirdly, the retrospective nature limits the generalizability of the study findings as selction bias cannot be completely accounted for. No formal sample size calculation available for determining the power of the study to draw these conclusions.

## CONCLUSIONS

The pathological process of atrial fibrosis can significantly increase the clot burden in patients with AIS induced by NVAF. Atrial fibrosis increases the risk and severity of stroke, making such patients more prone to thrombosis during an episode of AF and causing these clots to accumulate in the cerebrovascular system. Atrial fibrosis does not directly determine the physical and chemical properties of the thrombus, nor is it directly related to blood clot density. Therefore, in the treatment and prevention of this type of ischemic stroke, in addition to focusing on the management of atrial fibrosis, it is also necessary to consider other factors affecting the characteristics of thrombus in order to develop a more comprehensive and effective treatment strategy.

### Funding
The authors received no funding for this work.

### Competing Interests
The authors declare there are no competing interests.

### Author Contributions
- Juan Zhao conceived and designed the experiments, performed the experiments, analyzed the data, prepared figures and/or tables, authored or reviewed drafts of the article, and approved the final draft.
- Guangjun Deng conceived and designed the experiments, performed the experiments, prepared figures and/or tables, authored or reviewed drafts of the article, and approved the final draft.
- Weijing Wang conceived and designed the experiments, prepared figures and/or tables, authored or reviewed drafts of the article, and approved the final draft.
- Peng Wang analyzed the data, prepared figures and/or tables, and approved the final draft.
- Xinyu Shen analyzed the data, authored or reviewed drafts of the article, and approved the final draft.

- Xiaoxiao Yuan performed the experiments, prepared figures and/or tables, and approved the final draft.
- Haifei Jiang conceived and designed the experiments, performed the experiments, authored or reviewed drafts of the article, and approved the final draft.
- Zhong-bao Ruan conceived and designed the experiments, authored or reviewed drafts of the article, and approved the final draft.

## Human Ethics

The following information was supplied relating to ethical approvals (i.e., approving body and any reference numbers):

The Tongzhou People's Hospitalr granted Ethical approval to perform the study within its facilities (Ethical Application Ref: 2021-K012).

## Data Availability

The raw measurements are available in the Supplementary Files 1 and 2.

## Supplemental Information

Supplemental information for this article can be found online at http://dx.doi.org/10.7717/peerj.19173#supplemental-information.

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
