# Peer review of "Effect of atrial fibrosis on clot burden score and physicochemical properties of thrombus in patients with ischaemic stroke occurring in non-valvular atrial fibrillation"

_PeerJ, doi:10.7717/peerj.19173_

## Round 0.1 · original submission · Minor Revisions

Please address concerns of all reviewers and revise manuscript accordingly.

Reviewer 1 ·

Basic reporting

In this interesting paper, the authors analyzed the strong relationship between atrial fibrillation and acute ischemic stroke (AIS). This relationship is mediated by the atrial fibrosis.

The authors exhaustively described several electrocardiographic and laboratoristic biomarker of atrial fibrosis.

In the Discussion section, the authors remarked the important role exerted by inflammatory response in AIS patients in promoting atrial fibrosis and causing hypercoagulability.

However, they did not mention an innovative biomarker of left atrial fibrosis, that is the left atrial reservoir strain (LASr), assessed by speckle tracking echocardiography (STE).

STE examination should be considered for implementation in the clinical practice, due to its incremental diagnostic and prognostic value over conventional transthoracic echocardiography.

Experimental design

The experimental design is appropriate.
The methods section is adequate.

Validity of the findings

The authors' findings are interesting.
However, in the Discussion section the authors could also mention the usefulness of speckle tracking echocardiography for detecting myocardial fibrosis in AF patients with AIS.

Additional comments

In this interesting paper, the authors analyzed the strong relationship between atrial fibrillation and acute ischemic stroke (AIS). This relationship is mediated by the atrial fibrosis.

The authors exhaustively described several electrocardiographic and laboratoristic biomarker of atrial fibrosis.

In the Discussion section, the authors remarked the important role exerted by inflammatory response in AIS patients in promoting atrial fibrosis and causing hypercoagulability.

However, they did not mention an innovative biomarker of left atrial fibrosis, that is the left atrial reservoir strain (LASr), assessed by speckle tracking echocardiography (STE). The lower the LASr magnitude, the greater the extent of myocardial fibrosis on cardiac magnetic resonance imaging (PMID: 20133512).
In the Discussion section, on line 226, the authors could also report that recent evidence indicates that LASr impairment in AF patients may noninvasively predict left atrial appendage dysfunction/blood stasis/thrombosis.
STE examination should be considered for implementation in the clinical practice, due to its incremental diagnostic and prognostic value over conventional transthoracic echocardiography.

Reviewer 2 ·

Basic reporting

Clear use of English language. Nice study idea but there are limitations. References can be updated to include most recent literature Statistics are fine and the tables/images of decent quality. The manuscript can easily be made more concise and straightforward at many places as suggested. Please see the comments below for details.

Experimental design

There are concerns with the methodology which I have highlighted below in greater detail. some of them can be addressed and it will help improve the quality of the Manuscript. Please see the comments below for details.

Validity of the findings

The study is indeed innovative and novel to an extent. The data seems clean and statistical tools applied are also correct. Some aspects can be improved as suggested below:-

Additional comments

In the index report, the authors highlight the impact of the degree of atrial fibrosis on the clot burden score (CBS) and physicochemical properties in patients with acute ischaemic stroke (AIS) due to non-valvular atrial fibrillation (NVAF). The manuscript is timely, clinically relevant and of use to physicians, neurologists and cardiologist. I have minor concerns which need to be addressed: -
1) In the abstract, Please mention the study was retrospective in nature, remove the name of hospital from the abstract and the main text. Expand PTFV1 in abstract. Clearly indicate for authors that CBS 0-6 signifies higher thrombus burden, otherwise one may find it confusing and contradictory. Remove the statidtcal details at most places in the abstract and keep it simple, concise and relevant. Your abstract in the current state gives it a monotonous imporession
2) Expand AIS in introduction and add that AF is a risk factor for Coronary artery disease and ischemia and also dementia besides stroke and HF. For instance, one prospective observational study reported dementia in almost 37% of all NVAF patients. include these details and also provide the relevant recent references. Dementia in turn, is partly linked to atrial fibroses as seen in certain studies
3) The retrospective nature limits the generalizability of the study findings as selction bias cannot be completely accounted for. No formal sample size calculation available for determining the power of the study to draw these conclusions
4) Remove the details of (TOAST) classification from methods
5) How was PTFV1 calculated. Provide details and diagram of ECG along with the same for the readers understanding and to allow for repeatability of your findings in future by other studies.
6) Another major limitation is that one cannot assume that GAL-3 and TGF B will always be synonymous with atrial fibrosis. These markers are non-specific and in general would indicate excess inflammation and fibrosis in the body and not specifically localizing to the atrium. The only definitive evidence would come from atrial biopsy possible at the time of some other cardiac surgery (although it is not a realistic one). Perhaps using LA strain in combination with these biomarkers would make more sense and give a definitive alternative to biopsy for quantifying atrial fibroses
7) Mention mean age in results up to 2 decimals with standard deviation
8) The first line of discussion is a repeat. Please avoid
9) Please check this statement ‘Stroke caused by AF is usually fatal or results in severe disability.’ And give valid reference to it if it is appropriate
10) No need to include details of all the biomarkers like BNP, NLR etc. just stick to the most relevant ones like GAL-3 and TGF B will and make the discussion short, concise and more relevant to your research.
11) Another inflammatory marker besides the ones discussed and researched by the authors is initial or baseline Hs-CRP which in fact of late has been shown to independent predictor of clinical outcome in NVAF. Perhaps the authors can include 1-2 recent papers highlighting the same in the references for the readers
12) I would suggest the authors to further mention in discussion (although the same has not been studied in the index paper) that atrial fibroses has a strong link with underlying ischemia in the presence or absence of epicardial stenosis and this in turn is one of the strongest predictors of clinical outcomes including CAD, MI, stroke, HF, recurrence after ablation etc. accorindgly, I will suggest addition of some of the most relevant literature surrounding this published in the last 2 years.
13) The conclusions should be specific, focused and discussing the salient findings and key messages from your study rather than previously known facts. Make it concise and straightforward

---

## Round 0.2 · Minor Revisions

Please address remaining issues pointed by the reviewers and amend the manuscript accordingly.

Reviewer 1 ·

Basic reporting

The manuscript is considerably improved.

Experimental design

The experimental design is adequate.

Validity of the findings

The findings are interesting for clinical cardiologists.

Additional comments

In the Discussion section, in regard to the following sentence: "Recent evidence indicates that LASr impairment in AF patients may noninvasively predict left atrial appendage dysfunction and thrombosis", the authors could cite the following references: PMID: 24354939, PMID: 28559531 and PMID: 33389359.

Reviewer 2 ·

Basic reporting

please refer to detailed comments below

Experimental design

please refer to detailed comments below

Validity of the findings

please refer to detailed comments below

Additional comments

while majority of the concerns raised by the fellow reviewer and myself have adequately been addressed by the authors, I would urge them to better highlight the role of ischemia at microcirculatory level in producing heterogeneity in atrial conduction system thereby promoting AF. SO, I would suggest the authors to further mention in discussion (although the same has not been studied in the index paper) that atrial fibroses has a strong link with underlying ischemia in the presence or absence of epicardial stenosis and this in turn is one of the strongest predictors of clinical outcomes including CAD, MI, stroke, HF, recurrence after ablation etc. Accorindgly, I will suggest addition of some of the most relevant literature surrounding this published in the last 2 years. I would suggest addition of these papers published recently and highlighting the same: "Atrial fibrillation and coronary artery disease: An integrative review focusing on therapeutic implications of this relationship", "Assessment of Coronary Artery Disease in Non-Valvular Atrial Fibrillation: Is This Light at the End of the Tunnel?", "The relationship between atrial fibrillation and coronary artery disease: Understanding common denominators"

---

## Round 0.3 · accepted · Accept

All remaining concerns were addressed and revised manuscript is acceptable now.